# Population Dynamics in the Biogenesis of Single-/Multi-Layered Membrane Vesicles Revealed by Encapsulated GFP-Monitoring Analysis

**Sangho Koh** [1], **Shuhei Noda** [1,2] and **Seiichi Taguchi** [1,3,*]

[1] Graduate School of Science, Technology, and Innovation, Kobe University, 1-1 Rokkodai-cho, Nada, Kobe 657-8501, Japan; koh@port.kobe-u.ac.jp (S.K.); shuhei.noda@opal.kobe-u.ac.jp (S.N.)
[2] PRESTO, Japan Science and Technology Agency (JST), Saitama 332-0012, Japan
[3] Engineering Biology Research Center, Kobe University, 1-1 Rokkodai-cho, Nada, Kobe 657-8501, Japan
[*] Correspondence: staguchi86@people.kobe-u.ac.jp

**Abstract:** Various generations of membrane vesicles (MV) have been observed in *Escherichia coli* in terms of triggering events and populations of single-layered (s)/multi-layered (m) forms. Previously, we proposed a novel mechanism for MV generation triggered by the intracellular accumulation of biopolyester polyhydroxybutyrate (PHB). This was designated as the Polymer Intracellular Accumulation-triggered system for Membrane Vesicle Production (PIA-MVP). Herein, we attempted to determine the conditions for the change in the population between s-MV and m-MV using glucose concentration-dependent PIA-MVP. PIA-MVP was established using the good correlation between the glucose concentration-dependent PHB accumulation and MV generation. Thus, we assumed the presence of a critical glucose concentration could determine the population ratio of s-MV to m-MV, indicating that s-MV generation is a dominant component in the extracellular environment. Cytoplasmic green fluorescent protein (GFP) was used to evaluate the glucose concentration, enabling the selective generation of s-MV. The glucose concentration was determined to be 15 g/L to satisfy this purpose under the culture conditions. In conclusion, we established a biological system allowing us to selectively generate both single- and multi-layered MVs based on PIA-VIP encapsulation of GFP, providing a versatile toolkit to gain insights into the MV generation mechanism and achieve progress in various engineering applications.

**Keywords:** bacterial membrane vesicle; outer membrane vesicle; bilayer vesicle; multi-lamellar vesicle; polyhydroxybutyrate; biopolyester; protein secretion; fluorescent monitoring; GFP reporter; *Escherichia coli*

## 1. Introduction

The formation of naturally occurring membrane vesicles (MVs) has been observed in nature, and they can be defied as a dynamic morphological differentiation from the mother cells of any living organism after exposure to external environmental changes. Although this type of differentiation seems rare, we can speculate on the mechanisms of MV generation that have not been elucidated so far from the literature [1,2]. Basically, the biogenesis of MVs is a natural physiological phenomenon in prokaryotic and eukaryotic cells. These nano-sized lipid capsules are 20–400 nm in diameter and encapsulate cellular biomacromolecules, such as DNA, RNA, proteins, and small molecules [1]. In addition, bacterial MVs play a role in microbial cell–cell interactions, such as in horizontal gene transfer and quorum sensing. They can be found in many environmental and physiological sites. Various studies have reported on utilizing MVs for a wide range of applications, and understanding the mechanisms underlying MV generation is crucial when considering their physiological significance and any applications of interest [1].

Based on their double-layer membrane structures, gram-negative bacteria have two distinct types of extracellular MVs, single-layered MVs (s-MVs) and multi-layered MVs (m-MVs). The budding process primarily generates s-MVs from the outer membrane that encapsulate periplasmic compounds [2]. In contrast, m-MVs consist of both outer and inner membranes, allowing them to encapsulate both periplasmic and cytoplasmic compounds [3]. The possibility of encapsulation of cytoplasmic compounds into MVs was proposed in 1995 [4], although the existence of m-MVs was not yet confirmed at that time. However, recent studies using state-of-the-art microscopy techniques have shown that m-MVs with double lamellae or more complex structures are released from gram-negative bacteria [3]. In the case of gram-negative bacteria, m-MV generation occurs due to heightened envelope stress related to disruptions or imbalances in the membrane architecture [3]. There are two major mechanisms of m-MV formation. In the first mechanism, blebbed vesicles derived from internal membranes are trapped in the curvature of the outer membrane where the peptidoglycan (PG) is weakened. These events have been observed in glycine-induced PG-damaged cells of *Escherichia coli* [5]. In addition, this process is also observed in the hyper vesiculation mutant ΔmlaEΔnlpI, in which phospholipids accumulate in the outer leaflet of the outer membrane and the crosslinking between lipoprotein and PG is decreased [6,7]. Furthermore, m-MV formation was observed in the *E. coli* ΔtolA mutant in relation to Tol-Pal crosslinking, which connects the outer membrane, PG, and inner membrane via protein–PG and protein–protein interactions [8]. The second mechanism of m-MV formation involves endolysin-based explosive cell lysis, which is triggered by DNA damage [9]. In this process, DNA-damaging stress-induced PG degradation by endolysins causes cell lysis, resulting in broken outer and inner membranes being re-assemble as a vesicle. Notably, variations in triggering events and random biogenesis of s-MVs and/or m-MVs are frequently observed as a mixture of s-MV and m-MV, suggesting the fluctuating complexity of the vesiculation process. The frequency population ratio of the s-MV and m-MVs generated depends on the impact of the envelope stress, which is weaker for s-MV and stronger for m-MV. Importantly, no distinct selective/regulatory generation condition of either type of MV has been reported so far.

Recently, we discovered a novel MV generation mechanism in *E. coli* cells that is triggered by the intracellular accumulation of poly-3-hydroxybutyrate (PHB), namely a Polymer Intracellular Accumulation-triggered system for MV Production (PIA-MVP) [10]. In general, the microbial polyester PHB accumulates in the cytoplasm as a carbon and energy storage polymer in response to excess carbon availability in many microbial species. PHB is a promising carbon-neutral biodegradable substance for use as a bio-based plastic [11]. Further, a link between PHB accumulation and stress tolerance has been observed in some bacteria, showing that PHB has more biological roles than merely serving as a carbon and energy storage molecule [12]. It has been suggested that PHB metabolism acts as a regulatory mechanism optimizing carbon and energy flow in a bacterial cell [12]. The PHB synthesis pathway in relation to MV production from glucose in a recombinant *E. coli* harboring a *phbCAB* synthetic gene operon from the natural PHB producer *Ralstonia eutropha* is described in Figure 1. PHB is synthesized from acetyl-CoA by sequential enzymatic reactions comprising 3HB-CoA-precursor supplying steps catalyzed by β-ketothiolase (PhbA) and NADPH-dependent acetoacetyl-CoA reductase (PhbB), and a polymerizing step catalyzed by PHB synthase (PhbC) [13]. In this pathway, PHB accumulation can be regulated by varying glucose concentrations in the culture medium [14]. Interestingly, we found that MV production is also governed by intracellular PHB production, which can be finely controlled by varying glucose concentrations through PIA-MVP [10]. It has also been proposed that PHB production causes envelope stress, as indicated by increased cell enlargement [10]. The glucose concentration-dependent PHB production is a key trigger for cell enlargement and MV biogenesis. Therefore, we assumed that the internal pressure from PHB intracellular accumulation toward the cell membrane structure could lead to MV formation due to envelope stress. Furthermore, in the case of PIA-MVP, both s-MV

and m-MV were generated under conditions of high PHB accumulation (higher internal pressure); however, the underlying mechanism remains unknown.

Since m-MV is composed of both the outer and inner membranes of the wild-type strain of *E. coli* cells, we hypothesized that m-MV generation would occur under higher PHB accumulation conditions, indicating higher internal pressure conditions. Furthermore, we attempted to determine the "borderline" between the generation of majority s-MVs and the occurrence of m-MV by varying the glucose concentration of the media. For this purpose, we semi-quantitatively evaluated the population dynamics for the ratio of s-MV to m-MV using glucose-dependent PIA-MVP. Based on the principle of cellular architecture, we assumed that cytoplasmic green fluorescence protein (GFP) would specifically be encapsulated into m-MVs and could be used as a fluorescence reporter, whereas s-MVs would fail to encapsulate cytoplasmic GFP, as reported previously. In this study, we demonstrated the selective/regulatory generation of s-MVs as a dominant component below the critical glucose concentration by monitoring GFP intensity. To the best of our knowledge, the complete encapsulation of GFP as a free form into MVs has not yet been achieved. Therefore, the establishment of this technique presents a breakthrough in this field for investigating population dynamics between s-MV and m-MV. As a proof-of-concept, these results were finally confirmed by cross-sectional transmission electron microscopy (TEM).

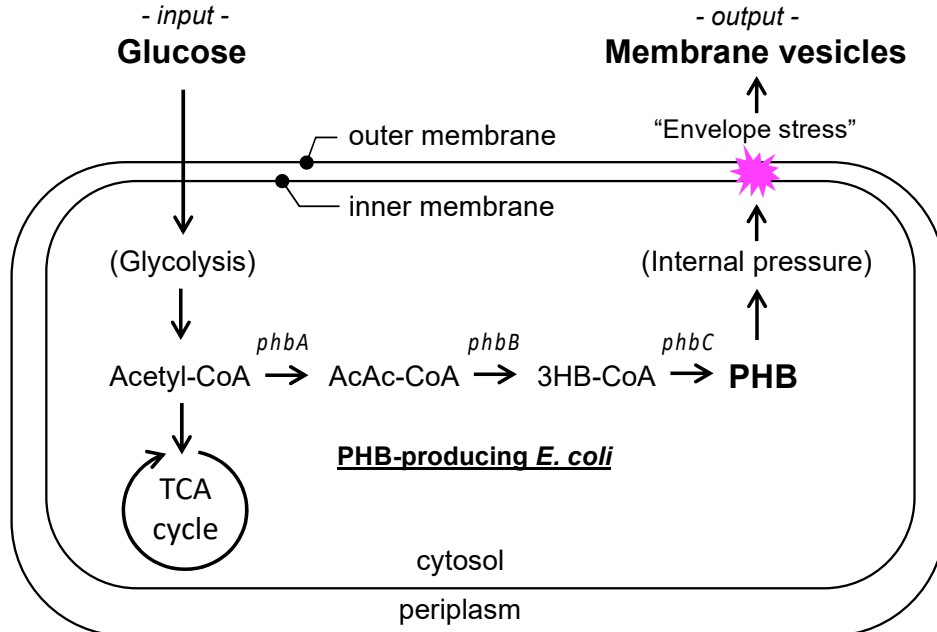

**Figure 1.** Polyhydroxybutyrate (PHB) (synthesis pathway in relation to membrane vesicle (MV) production). Acetyl-CoA (Ac-CoA) supplied from glucose via glycolysis is converted to acetoacetyl-CoA (AcAc-CoA) by heterologously expressed PhbA. AcAc-CoA is converted to 3-hydroxybutyrate-CoA (3HB-CoA) by PhbB. 3HB-CoA is utilized as a monomer for the polymerization by PhbC.

## 2. Materials and Methods

### 2.1. Bacterial Strains and Culture Conditions

*E. coli* strain BW25113 [15] obtained from the National Institute of Genetics, Japan, was used as the host strain. The plasmid pGEM-phaC$_{Re}$AB [10] carrying the PHA synthase gene (phaC$_{Re}$ from *R. eutropha*), 3-ketothiolase gene (phaA$_{Re}$ from *R. eutropha*), and acetoacetyl-CoA reductase gene (phaB$_{Re}$ from *R. eutropha*) was used for PHB production. The plasmid pCA24N-gfp [15], carrying the GFP gene under the control of the lac promoter, was introduced into a recombinant *E. coli* strain harboring pGEM-phaC$_{Re}$AB. The recombinant *E. coli* was cultured in Luria Bertani (LB) medium (10 g/L tryptone, 5 g/L yeast extract, and 10 g/L NaCl) containing 20 g/L glucose with 100 mg/L ampicillin, 50 mg/L chlorampheni-

col, and 0.1 mM isopropyl β-D-thiogalactopyranoside (IPTG) for the production of PHB and GFP. When glucose-dependent PHB production was examined, its concentration varied as follows: 0, 5, 10, 15, and 20 g/L. The culture medium for the strains harboring plasmid vectors pGEM-phaC$_{Re}$AB and pCA24N-gfp was supplemented with 100 mg/L ampicillin, 50 mg/L chloramphenicol, and 0.1 mM IPTG. All the test cultures were precultured in LB for 7 h at 30 °C and inoculated into 100 mL of fresh LB in a 500 mL Sakaguchi flask. Recombinant strains were cultivated by reciprocal shaking at 125 strokes/min for 48 h.

## 2.2. Quantification of the Intracellular PHB Accumulation

After cultivation, the collected cells were washed three times with water and freeze-dried. The PHB content in the cells was determined via gas chromatography (GC) using methanolized dried cell samples, as previously described [10]. Approximately 50 mg of dried cells was briefly methanolized in 2 mL of 15:85% (*v/v*) sulfuric acid/methanol and 2 mL of chloroform at 100 °C for 140 min. After cooling, 1 mL of pure water was added and the mixture was left until it separated into two layers. A portion of the chloroform layer was subjected to GC (GC-2030, Shimadzu, Kyoto, Japan) equipped with a flame ionization detector (FID), and methyl-esterified 3HB derived from P(3HB) was detected.

## 2.3. Quantification of Extracellular MV Production and GFP Encupsulation into MVs

MVs were isolated and quantified as described previously with some modifications [10]. Briefly, 20 mL of culture supernatant was filtered with a 0.45 μm pore size cellulose acetate filter (Merck Millipore, Billerica, Massachusetts, USA) and ultracentrifuged for 1 h at 150,000× *g* and 4 °C. The pellets were resuspended in double-distilled water for MV quantification. For further purification, MV pellets were resuspended in 45% iodixanol (Optiprep, AXIS-SHIELD, Dundee, UK) in 10 mM HEPES/0.85% NaCl buffer and subjected to density gradient ultracentrifugation for 3 h at 100,000× *g* and 4 °C in an iodixanol gradient of 40, 35, 30, 25, and 20%; 200 μL per each fraction were taken from the top.

For MV quantification, isolated MVs were dissolved in 200 μL of phosphate-buffered saline (PBS) and incubated with 1.25 μg/mL FM4-64 (Molecular Probes, Invitrogen, Carlsbad, CA, USA) in PBS. Thereafter, MVs were measured at excitation and emission wavelengths of 558 and 734 nm, respectively, using the microplate reader Synergy H1 (BioTek Instruments, Winooski, VT, USA). The MV sample without staining by FM4-64 was used as the negative control. GFP capsulation was also quantified via fluorescence measurements. The fluorescence intensity of GFP in MVs was measured at excitation and emission wavelengths of 395 and 509 nm, respectively.

## 2.4. Cross-Sectional TEM Analysis

Culture samples of the recombinant *E. coli* or purified MVs were fixed with 2.5% glutaraldehyde in 10 mM PBS and stored at 4 °C for a few days. Subsequently, the sample was rinsed twice with 10 mM PBS buffer, post-fixed with 2% osmium tetroxide, and dehydrated with increasing concentrations of ethanol as follows: 50, 70, 90, and 99%. The dehydrated sample was then embedded in the epoxy resin Quetol 651 (Nisshin EM Co., Ltd., Tokyo, Japan). The resulting epoxy resin block was then placed on an Ultracut S ultramicrotome (Leica, Wetzlar, Germany) for sectioning by cutting with a diamond knife at an angle of 45°. Selected 50–60 nm ultra-thin sections were placed on copper EM F-200 grids (Nisshin EM Co., Ltd.) with 200 mesh size and stained with 2% uranyl acetate. The resulting ultrathin sections were observed under a TEM (JEM-2010) at 120 kV (JEOL, Tokyo, Japan).

## 3. Results and Discussion

### 3.1. Working Hypothesis of PIA-MVP as a Regulatory Secretion Machinery of m-MVs

To understand the underlying mechanism of PIA-MVP, we focused on generating two types of MVs in *E. coli*, s-MVs and m-MVs. In fact, the internal layered-structure analysis of MVs using cross-sectional TEM demonstrated that bi-layered MVs budded and were released from *E. coli* cells containing 86 wt% (cellular contents) PHB when cultured in

20 g/L glucose (Figure 2). Using this observation, based on the population ratio of both s-MVs and m-MVs, it could be possible to evaluate the degree of envelope stress occurring in an artificial PHB intracellular accumulation system of *E. coli* bearing two distinct outer and inner membranes. We hypothesized that s-MVs would be generated even at lower levels of PHB accumulation (lower internal pressure). In contrast, m-MV biogenesis begins at a higher PHB accumulation level (higher internal pressure). Therefore, the relative ratio of s-MVs to m-MVs would be higher when the PHB production can be controlled at a lower level. In contrast, the proportion of m-MVs would increase with increasing PHB production. In fact, a direct correlation between the production levels of MV and PHB has been reported [10]; however, the population ratio of s-MV and m-MV is not clear. Thus, we assumed that the modes of m-MV generation were dependent on the degree of internal pressure controlled by the PHB accumulation level. This implies that the glucose concentration can be considered as a controllable stress effector for the generation of MVs in different modes, consequently regulating the population ratios of s-MVs and m-MVs. Based on this hypothesis, we attempted to determine the optimum conditions for the regulatory secretion of m-MVs by varying glucose concentrations.

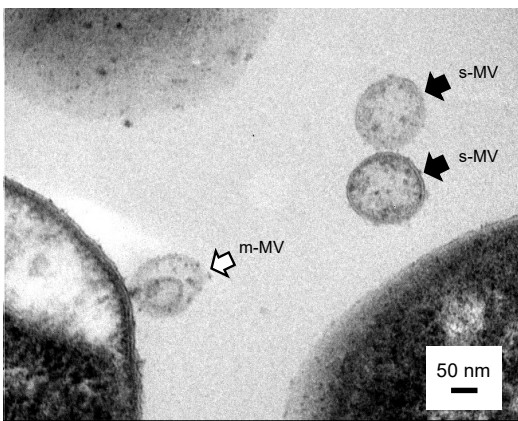

**Figure 2.** Transmission electron microscopy (TEM) micrographs of cross-sections of polyhydroxy-butyrate (PHB)-producing *Escherichia coli.* Recombinant *E. coli* cells intracellularly producing PHB and green fluorescence protein (GFP) were collected after 48 h of incubation with 20 g/L glucose. The black arrowhead indicates the single-layered MV (s-MV). The open arrowhead indicates the multi-layered MV (m-MV). Scale bars, 50 nm.

### 3.2. Direct Evidence of Encapsulation of GFP Reporter Protein into MVs

To evaluate the modes of generation of s-MVs and m-MVs, we investigated the ability of MV to encapsulate cytoplasmic GFP reporter proteins. Thus, GFP was expressed in PHB-producing *E. coli* and was cultivated with 20 g/L glucose, which is routinely used for PHB fermentation. As shown in Figure 3, density gradient centrifugation demonstrated that the MV-containing fraction purified from the culture supernatant of the recombinant strain clearly showed GFP-derived fluorescence. In addition, a dense accumulation of GFP was observed via fluorescent microscopy (Figure 3b), suggesting that cytoplasmic GFP was efficiently encapsulated in the MV. These results suggest the potential establishment of a versatile MV secretion machinery capable of transporting difficult-to-secrete cytoplasmic polymers, such as heterologous value-added products, into the extracellular environment.

To the best of our knowledge, this is the first direct evidence of cytoplasmic GFP inclusion in MVs. The use of periplasmic GFP in MV studies has been previously reported [16,17]. However, encapsulation of periplasmic GFP into MVs was mostly revealed via immunoblot analysis using an anti-GFP antibody. These results implied that large cytoplasmic GFP molecules could not be readily transported to the MV cargo. It should be noted that PIA-MVP is a promising secretion machinery for the efficient encapsulation of cytoplasmic GFP into MVs when compared with the previously reported MV generation modes. Furthermore, PIA-MVP was produced in *E. coli* without any stress-limited

mutations in the cell membrane architecture. Thus, PIA-MVP is a versatile methodology for efficient MV production that can be combined with conventional triggers for MV biogenesis, such as imbalances in the outer membrane leaflet [12], mutations in the Tol-Pal system that maintains outer membrane integrity [4], and external addition of chemicals, such as glycine [13]. In particular, such an original MV-based secretion machinery could be a powerful add-on to the immature secretion system of *E. coli*.

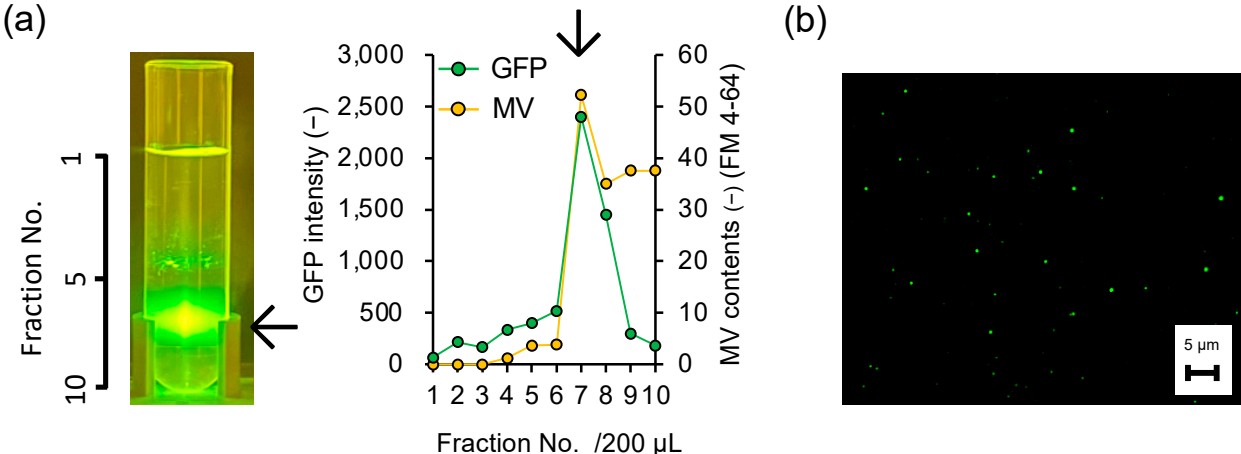

**Figure 3.** Direct evidence of encapsulation of GFP into MV. (**a**) Purification of extracellular MVs by density-gradient ultracentrifugation. A black arrowhead indicates the peak that is synchronized with both MV contents and GFP intensity. (**b**) Fluorescence microscopic image of purified MVs encapsulating GFP. Scale bars, 5 μm.

### 3.3. Is the Selective/regulatory Secretion of s-MV Possible?

Next, we focused on the possibility of the regulatory secretion of s-MVs and m-MVs at a tailor-made ratio. Based on the principle of PIA-MVP, we hypothesized that the population ratio between s-MVs and m-MVs depends on the impact of envelope stress, which is weaker for s-MVs and stronger for m-MVs. Therefore, we evaluated whether the population ratio between s-MVs and m-MVs was altered depending on the PHB accumulation level, which was controlled by varying the glucose concentration. Considering the cellular architecture, these cytoplasmic compounds should be specifically encapsulated in m-MVs, including bi-layered MVs, whereas s-MVs should specifically encapsulate periplasmic compounds and not encapsulate GFP. As shown in Figure 4a,b,d, a linear correlation was observed between the production levels of MVs and PHB. In contrast, no GFP-derived fluorescence was observed at glucose concentrations of 0–15 g/L, whereas GFP-derived fluorescence was observed at concentrations over 15 g/L (cellular PHB 68 wt%) (Figure 4c,e). It is likely that MVs generated below this condition contain s-MVs as the major population with no/very low recruitment of GFP, which is more readily packaged into m-MVs. Thus, the glucose concentration of 15 g/L was determined to be the borderline beyond which s-MV production would no longer be the majority MV type in the population.

Finally, cross-sectional TEM analysis was performed to compare the internal layered structures of the two distinct MVs generated under lower and higher glucose concentrations (5 g/L and 20 g/L). As shown in Figure 5, s-MVs were the major population present at lower glucose concentrations. In contrast, the appearance of s-MVs in addition to three major m-MVs, including double bilayer type, multilamellar type (≥3 lipid bilayers), and grouped intravesicular type, was observed as a mixture at 20 g/L glucose concentration, implying that the inner membrane containing m-MVs mainly occurred under higher PHB-accumulation conditions. These results suggest the presence of a critical point that determines the population ratio of s-MVs to m-MVs. These results suggest a borderline glucose concentration that can selectively generate the s-MV major fraction, the mixed fraction of s-MV and m-MV, and the m-MV major fraction.

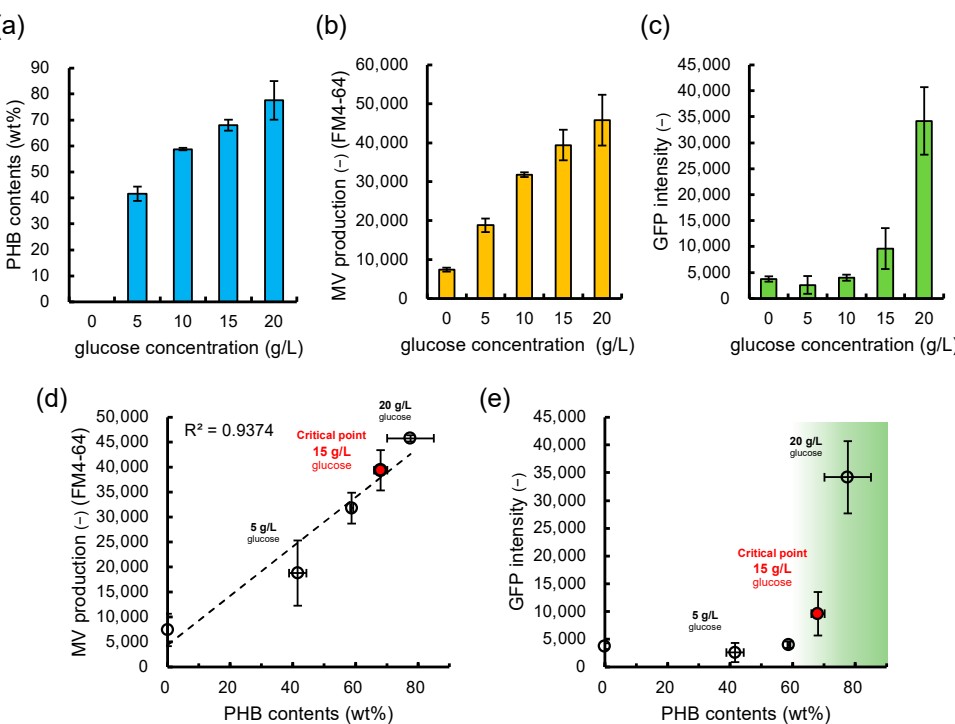

**Figure 4.** Relationship between MV production, GFP intensity in MVs, and intracellular PHB contents in the recombinant *E. coli*. intracellular PHB contents (**a**), extracellular MV production (**b**), and GFP intensity (**c**) in purified MVs from the recombinant *E. coli* under five different glucose concentrations (0 g/L, 5 g/L, 10 g/L, 15 g/L, and 20 g/L). (**d**) There is a straight correlation between the PHB contents and MV production level ($R^2$ = 0.9374). (**e**) There is no correlation between GFP intensity and PHB contents, while GFP intensity was observed over 15 g/L (cellular PHB 68 ± 2 wt%) as indicated by the red plot. The green zone shown in subfigure (**e**) is an illustrative representation intended to depict the occurance of GFP encupsulation. The data are shown as the mean ± standard deviation from three independent experiments.

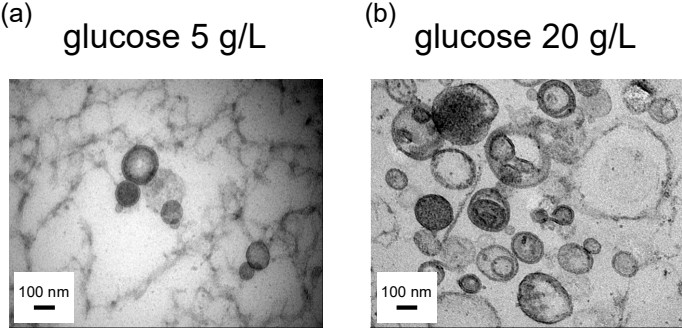

**Figure 5.** TEM micrographs of cross-sections of two distinct types of MVs generated under the following glucose concentrations: (**a**) 5 g/L; (**b**) 20 g/L. (**a**) s-MV was observed as the major population at 5 g/L glucose condition. (**b**) m-MV was frequently observed when the PHB production was controlled at a higher level under the 20 g/L glucose concentration. Scale bars, 100 nm.

### 3.4. Significance and Implication

In *E. coli*, the presence of "envelope stress" in the membrane architecture has been well established for MV production [2]. The envelope stress is caused by a disorder of the linkage network between three structured biomacromolecules, namely the outer membrane, the PG layer, and the inner membrane. Several reports have shown that mutants lacking genes related to outer membrane construction and encoding lipoprotein and PG-induced vesicle production [6–8,18]. Accordingly, these mutations induced random generation

of s-MVs or m-MVs depending on the given physiological situations, with variations in triggering events, suggesting the fluctuating complexity of the vesiculation process.

Conversely, PIA-MVP has provided a different approach in which the trigger for MV formation is PHB accumulation itself. Furthermore, this study sought to address the uniqueness of this system and the mechanisms commonly shared with other systems. Therefore, we focused on the population ratio of s-MVs (outer MVs) and m-MVs, including double-layered MVs (outer/inner MVs) that are generated as a mixture upon PHB accumulation. Assuming a working hypothesis that the ratios of both MVs would be regulated by glucose concentration-dependent PHB accumulation level, we confirmed the encapsulation of GFP in our model.

Here, the visualization approach using the GFP system was used for the monitoring of the population dynamics in the biogenesis of s-MV and m-MV. As illustrated in Figure 6, the critical glucose concentration was determined to be 15 g/L (cellular PHB 68 wt%), indicating that the biogenesis of m-MVs was regulated by the glucose concentration. Thus, it might be possible that the biogenesis of m-MVs starts when a certain critical point of internal pressure, the envelope stress induced by PHB accumulation, is reached. This should be a characteristic feature that is distinguishable from those of other reported systems. It is well known that internalized vesicles in m-MVs can contain cytoplasmic compounds [3]. The vesiculation of cytoplasmic GFP prompted us to establish a versatile MV secretion machinery that can function as an extracellular transporter of periplasmic compound-encapsulated s-MVs and efficient secretion system for difficult-to-secrete cytoplasmic compounds by m-MVs. The successful GFP-monitoring is a game-changing technology in the MV research field and a can be employed as a powerful approach for in-depth investigations of the MV generation and its practical application.

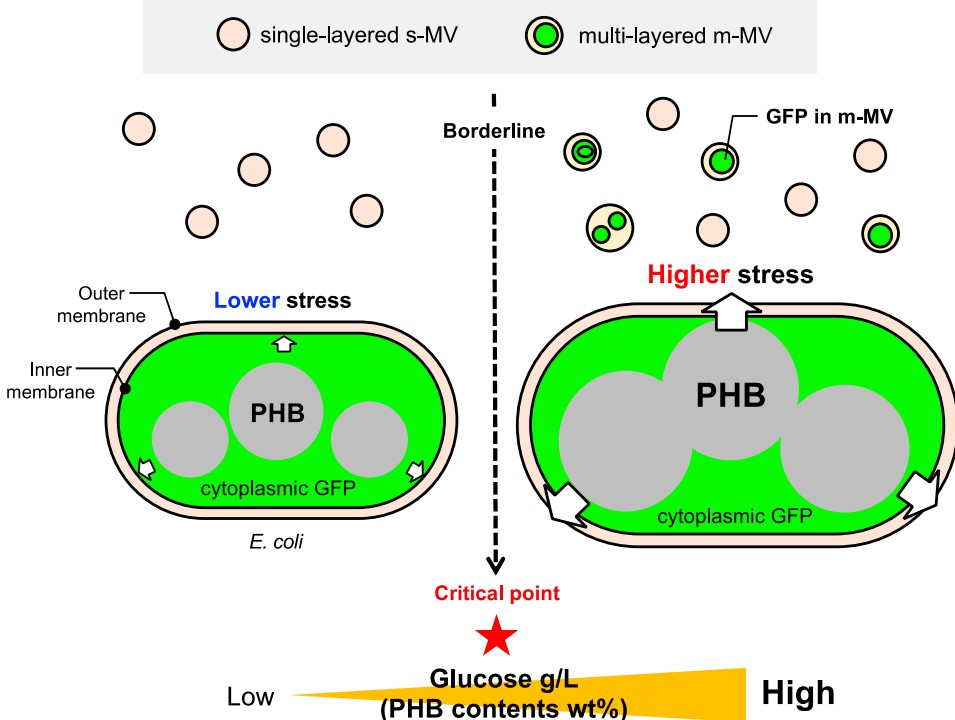

**Figure 6.** Proposed model of the glucose concentration-regulated population dynamics in the biogenesis of s-MV and m-MV by PIA-MVP. Based on the results of the GFP reporter experiments, a borderline glucose concentration can selectively generate the s-MV major fraction and a mixed fraction of s-MVs and m-MVs. The population ratio of s-MVs (caused by lower stress) to m-MVs (caused by higher stress) appeared to be inversely correlated with the increase in internal pressure caused by PHB intracellular accumulation. This means glucose concentration can be considered a controllable stress effector for generating MVs in a different mode, consequently regulating the population ratio of s-MVs and m-MVs.

On the other hand, MV biogenesis raises the fundamental issue of whether there is a physiological consequence of the intracellular accumulation of PHB in *E. coli*. The artificial *E. coli* system lacks the native carbonosome surface present in PHB-producing native bacteria like *R. eutropha* or *Pseudomonas putida*. Hence, the hydrophobic surface of the PHB produced in *E. coli* is at least partially exposed to the cytoplasm where it might come into contact with the cytoplasmic membrane, in particular because PHB granules in *E. coli* tend to localize close to the cell poles [19]. Throughout the cell cycle, physical contact of PHB with the cytoplasmic membrane was observed from the view of the electron-translucent structures. In addition, PHB granules produced by *Caryophanon latum* were associated with MVs in some frequency [20]. These natural ecosystems can provide insight into our artificial PIA-MVP system.

To date, many researchers have focused on microbial polyester PHB accumulation in bacterial cells because it can be used as biodegradable bio-based plastic. MVs are also well-known macromolecules that can be used for a wide range of applications, such as for vaccine display, in biofilm formation, and as drug delivery carriers. Our PIA-MVP system can greatly impact research in both PHB and MV and can also prompt researchers to carefully investigate a similar phenomenon in other biologically relevant species.

## 4. Conclusions

In this study, we demonstrated a proof-of-concept of the selective glucose concentration-regulated secretion of s-MVs by PIA-MVP. This suggests that PIA-MVP can serve as a versatile machinery for the selective secretion of periplasmic compound-encapsulated s-MVs and efficient secretion of difficult-to-secrete cytoplasmic compounds by encapsulated m-MVs. Notably, the selective secretion of s-MVs as a major population can be achieved by setting the glucose concentration below 15 g/L. From the fine encapsulation of cytoplasmic GFP into m-MVs and the sharp regulatory secretion of s-MVs, PIA-MVP-based MV secretion provides a promising platform for engineering applications and mechanistic studies. Here, we would like to emphasize that efficient encapsulation of GFP into m-MVs allowed us to visualize m-MV production. The present MV-based regulatory secretion machinery for production of value-added products can be upgraded in response to the required applications.

**Author Contributions:** S.K. and S.T. designed the experiments; S.K. conducted the experiments, S.K. and S.T. analyzed the results; S.K., S.N., and S.T. share the internal discussion; and S.K. and S.T. wrote the manuscript. All authors have read and agreed to the published version of the manuscript.

**Funding:** This research was funded by JSPS KAKENHI (23K13870 to S.K. and 19K22069 to S.T.).

**Institutional Review Board Statement:** Not applicable.

**Informed Consent Statement:** Not applicable.

**Data Availability Statement:** All data generated or analyzed during this study are included in this published article.

**Acknowledgments:** We would like to thank Michio Sato (Meiji University) for his support in the TEM analysis, Yoshihiro Ojima (Osaka Metropolitan University) for providing the plasmid pCA24N-gfp, and Ayaka Ojima in our laboratory for technical assistances.

**Conflicts of Interest:** The authors declare no conflict of interest.

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
