# Peer review of "Population Dynamics in the Biogenesis of Single-/Multi-Layered Membrane Vesicles Revealed by Encapsulated GFP-Monitoring Analysis"

_2673-8007, doi:10.3390/applmicrobiol3030070_

Round 1
Reviewer 1 Report
The research conducted by Taguchi and colleagues showcases an innovative method for glucose-triggered selective secretion of single-layered membrane vesicles (s-MVs) using the PIA-MVP mechanism. This advancement allows precise encapsulation and efficient secretion of specific compounds. While the paper's organization is commendable, some aspects could benefit from clarification:
1. Significance and Implications: To enhance comprehension of MV generation mechanisms and their implications, it would be valuable to elaborate on the broader significance of the findings. Detailing how these findings contribute to our understanding of cellular processes and their potential applications could provide readers with a more comprehensive perspective.
2. References for Envelope Stress Impact: To substantiate the claim about the varying impacts of envelope stress on s-MVs and m-MVs, including references that support this assertion would enhance the credibility and validity of your statement.
3. Fluorescence Differentiation in m-MVs: Addressing the challenge of distinguishing between m-MVs and other cellular compartments, such as the periplasmic and cytoplasmic regions, would shed light on how to precisely identify GFP localization within m-MVs, enhancing the accuracy of your observations.
Author Response
Responses to the comments by reviewer 1:
We would like to thank you for your positive and constructive comments to improve our manuscript. We have revised the manuscript accordingly.
- Significance and Implications: To enhance comprehension of MV generation mechanisms and their implications, it would be valuable to elaborate on the broader significance of the findings. Detailing how these findings contribute to our understanding of cellular processes and their potential applications could provide readers with a more comprehensive perspective.
>>>According to your suggestion, we have introduced a new section entitled “3.4 Significance and Implications” in the “Results and Discussion”. In this section, we provide a summary of how our findings contribute to applications and physiological aspects.
(See page 8, line 278-327)
3.4. Significance and Implication
In case of E. coli, an “envelope stress” in the membrane architecture has been well established for MV production [2]. The envelope stress is caused by disorder of the linkage network between three structured biomacromolecules, namely outer membrane, PG, and inner membrane. Several reports have shown that mutants lacking genes related to outer membrane construction and encoding lipoprotein and PG induce vesicle production [2]. Accordingly, these mutations induced random generation of s-MVs or m-MVs depending on the given physiological situations, with variations in triggering events, suggesting the fluctuating complexity of the vesiculation process.
Conversely, PIA-MVP has provided a different approach in which the trigger for MV formation is PHB accumulation itself. Furthermore, we were interested in addressing the uniqueness of this system and the mechanisms commonly shared with other systems. Therefore, we focused on the population ratio of s-MVs (outer MVs) and m-MVs, including double-layered MVs (outer/inner MVs) that are generated as a mixture upon PHB ac-cumulation. Assuming a working hypothesis that the ratios of both MVs would be reg-ulated by glucose concentration-dependent PHB accumulation level, we confirmed the possibility of encapsulation of GFP.
Here, the visualization approach with aid of GFP system should be a determinant for monitoring of the population dynamics in the biogenesis of s-MV and m-MV. As illustrated in Figure 6, the critical glucose concentration was determined to be 15 g/L (cellular PHB 68 wt%), indicating that the biogenesis of m-MVs was regulated by varying the glucose concentration. Thus, it might be considered that the biogenesis of m-MVs starts when a certain critical point of internal pressure, as an “envelope stress” induced by PHB accumulation, is reached. This should be a characteristic feature that is distinguishable from those of other reported systems. It is well known that internalized vesicles in m-MVs can contain cytoplasmic compounds [3]. The vesiculation of cytoplasmic GFP prompted us to establish a versatile MV secretion machinery that can function as an extracellular transporter of periplasmic compound-encapsulated s-MVs and efficient secretion system for difficult-to-secrete cytoplasmic compounds by m-MVs as an encapsulated form. The successful GFP-monitoring is a game-changing technology in the MV research field and a powerful approach for in-depth investigation of the MV generation and it's practical application.
On the other hand, MV biogenesis raises the fundamental issue of whether there is a physiological consequence of the intracellular accumulation of PHB in E. coli. The artificial E. coli system lacks the native carbonosome surface present in PHB-producing native bacteria like R. eutropha or Pseudomonas putida. Hence, the hydrophobic surface of the PHB produced in E. coli is at least partially exposed to the cytoplasm where it might come in contact with the cytoplasmic membrane, in particular because PHB granules in E. coli tend to localize close to the cell poles [19]. Throughout the cell cycle, physical contact of PHB with the cytoplasmic membrane was observed from the view of the electron-translucent structures. In addition, PHB granules produced by Caryophanon latum were associated with MVs in some frequent [20]. These natural ecosystems would provide any insights into our artificial system PIA-MVP.
To date, many researchers have focused on the most-studied microbial polyester PHB accumulated in bacterial cells because it can be used as biodegradable bio-based plastic. MVs are also well-known macromolecules that can be used for a wide range of applications, such as vaccine display, biofilm formation, and drug delivery carrier. Our PIA-MVP will have a great effect on both research fields of the popular macromolecules, PHB and MVs, and will also prompt PHA researchers to carefully investigate a similar phenomenon.
- References for Envelope Stress Impact: To substantiate the claim about the varying impacts of envelope stress on s-MVs and m-MVs, including references that support this assertion would enhance the credibility and validity of your statement.
>>>According to your suggestion, we have provided a comprehensive description of the concept of “envelope stress” in MV biogenesis in the “Introduction” section, and provided the relevant references.
(See page 1, line 32-36)
Nowadays, naturally occurring membrane vesicles (MVs) are encountered, which can be considered as a dynamic morphological differentiation originated by the mother cells of any living organism upon experiencing various external environmental changes. Although this type of differentiation seems rare, we can speculate on several mechanisms of MV generation that have not been elucidated so far from the increasing literature [1, 2].
(See page 1, line 40-45)
In addition, bacterial MVs play a role in microbial cell-cell interactions, such as in hori-zontal gene transfer and quorum sensing. They can be found in many environmental and physiological sites regardless of whether those sites are closed or open. Abundant in-formation has been accumulated on utilizing MVs for a wide range of applications, and understanding the mechanisms underlying MV generation is crucial when considering their physiological significance and any applications of interest.
(See page 2, line 49-69)
After that, in contrast, it was reported that m-MVs consist of outer and inner membranes, allowing them to encapsulate periplasmic and also cytoplasmic compounds [3]. The possibility of encapsulation of cytoplasmic compounds into MVs was proposed in 1995 [4], although the existence of m-MVs was not yet confirmed at that time. However, recent studies using state-of-the-art microscopy techniques have shown that m-MVs with double lamellae or more complex structures are released from gram-negative bacteria. In the case of gram-negative bacteria, m-MV generation occurs due to much more envelope stress related to disruptions or imbalances in the membrane architecture [3]. There are two major mechanisms of m-MV formation. In the first mechanism, blebbed vesicles derived from internal membranes are trapped in the curvature of the outer membrane where the peptidoglycan (PG) is weakened. Such an event was observed in PG-damaged cells of Escherichia coli induced by external glycine addition [5]. In addition, this process is ob-served in the hyper vesiculation mutant ΔmlaEΔnlpI, in which phospholipids accumulate in the outer leaflet of the outer membrane and the crosslinking between lipoprotein and PG is decreased [6,7]. Furthermore, m-MV formation was observed in the E. coli ΔtolA mutant in relation to Tol-Pal crosslinking, which connects the outer membrane, PG, and inner membrane via protein-PG and protein-protein interactions [8]. The second mechanism of m-MV formation involves endolysin-based explosive cell lysis, which is triggered by DNA damage [9]. In this process, DNA-damaging stress-induced PG degradation by endolysins causes cell lysis, resulting in broken outer membrane and inner membrane being re-assemble as a vesicle.
(See page 2, line 97-100)
The glucose concentration-dependent PHB production is a key trigger for cell enlarge-ment and MV biogenesis. Therefore, we assumed that the internal pressure from PHB intracellular accumulation toward the cell membrane architecture probably leads to MV formation due to envelope stress.
- Fluorescence Differentiation in m-MVs: Addressing the challenge of distinguishing between m-MVs and other cellular compartments, such as the periplasmic and cytoplasmic regions, would shed light on how to precisely identify GFP localization within m-MVs, enhancing the accuracy of your observations.
>>> Thank you for suggestion. At this moment, it is difficult to determine GFP localization. In the near future, to identify s-MV by a simple method, we will assess the periplasmic expression of RFP (Red Fluorescent Protein) as a marker.
Reviewer 2 Report
The authors presented a important work that has consequences in high-value product production through vesiculation of E.coli using low-value starting material. This is relevant work for this journal and I advice the editor to consider the manuscript for publishing after the below mentioned revisions.
Major concerns:
- The introduction can be improved to incorporate elements of how glucose is converted to PHB. The manuscript is lacking this crucial information.
- A more detailed mechanism of how PHB result in vesiculation needs to stated using existing literature or speculation in the introduction, apart from reiterating the idea of vesiculation under cytosolic stress.
- Upon referring the authors previous work - Ref no. 6 - under similar experimental conditions why is that MVs production is doubled in Fig 3b when compared to Fig 5a of Ref no.6 . Is this because of improved vesicle purification methods? Moreover, MV purification and quantification method section is missing!
- In Fig 1 - the bottom s-MV appears to have two contrasted layers. Are you sure it is s-MV? How do you define m-MV? - Those vesicles with a smaller internal vesicles in an s-MV? This clarification must be given. Moreover, the proportion of m-MVs to s-MVs in Figure 1 is 1:2 which is a misleading. I suggest the authors to employ more accurate depiction of their results in Fig 1. (higher m-MVs compared to s-MVs).
- A graph depicting the percentage of s-MVs and m-MVs as observed in TEM images or quantified through other means (for example, drawing conclusions from Fig 3b and Fig 3c) for 5% and 20% glucose conc. should be added to the figure 4. I believe without this information - the title - "selective generation" would not be accurate.
Minor corrections:
- Line 110 - sl-MV should be changed to s-MV
- Line 132 - Figure 2a instead of Figure 2
- Line 135 - "dense state of nanosized MV particles" - please explain
- Line 176 - It might be Figure 4 instead of Figure 5. Entire Figure 4 is missing from the text.
Author Response
Responses to the comments by reviewer 2:
We would like to thank you for your positive and constructive comments to improve our manuscript. We have revised the manuscript accordingly.
- The introduction can be improved to incorporate elements of how glucose is converted to PHB. The manuscript is lacking this crucial information.
>>> According to your suggestion, we have added a comprehensive description about PHB synthesis in the “Introduction” section. In addition, we have added a new figure, “Figure 1,” which illustrates the relationship between both PHB and MVs.
(See page 2, line 79-93)
In general, the microbial polyester PHB is intracellularly accumulated as a carbon and energy storage polymer in response to the availability of excess carbon source by many microbial species and is a promising carbon-neutral biodegradable substance for use as a bio-based plastic [11]. Further, a link between PHB accumulation and stress tolerance has been observed in some bacteria, showing that PHB has more biological roles than merely serving as a carbon and energy storage molecule [12]. Thus, there are many suggestions that PHA metabolism acts as a regulatory mechanism optimizing carbon and energy flow in a natural bacterial cell. The PHB synthesis pathway in relation to MV production from glucose in a recombinant E. coli harboring a phbCAB synthetic gene operon from the natural PHB producer Ralstonia eutropha is described in Figure 1. PHB is synthesized from acetyl-CoA by sequential enzymatic reactions comprising 3HB-CoA-precursor supplying steps catalyzed by β-ketothiolase (PhbA) and NADPH-dependent acetoacetyl-CoA re-ductase (PhbB), and a polymerizing step catalyzed by PHB synthase (PhbC) [13]. It is a well-known major pathway in which PHB accumulation level can be regulated by var-ying glucose concentrations in the culture medium [14].
Round 2
Reviewer 2 Report
Minor corrections:
- Section 3.3 - Mistake in section title
- Line 252, instead of under using the word 'below' is more meaningful.
- Figure 6 - Adjust the figure contents, many words are overlapped or not visible.
Author Response
Dear Prof. Dr. Ian Connerton
Special Issue Editor Applied Microbiology
Ms. Ruth Lin
Assistant Editor Applied Microbiology
We appreciate the constructive comments provided by the reviewer 2. These comments have helped improve the manuscript. Our point-by-point responses to the comments of the reviewer’s are also detailed below.
Also, we have two concerns for publishing our MS in Applied Microbiology.
- Could you please provide the title “Population Dynamics in the Biogenesis of Single-/Multi-Layered Membrane Vesicles Revealed by Encapsulated GFP-Monitoring Analysis” which revised after first-revision process.
- According to the reviewer’s advice and suggestions, we have revised the manuscript, and the word count of the main text is now > “4000 words”, which corresponds to the word count of an “Article”. In addition, we believe that the present version should include a full demonstration of the novel PIA-MVP-based mechanism enabling selective secretion of the target layered MVs other than only the results. Therefore, please reconsider our manuscript as a regular “Article” in Applied Microbiology.
Responses to the comments by reviewer 2:
- Section 3.3 - Mistake in section title
>>>According to your suggestion, we have revised the manuscript.
(See page 6, line 236) Is the Selective Secretion of s-MV Possible?
- Line 252, instead of under using the word 'below' is more meaningful.
>>>According to your suggestion, we have revised the manuscript.
(See page 6, line 249) “below”
- Figure 6 - Adjust the figure contents, many words are overlapped or not visible.
>>>According to your suggestion, we have revised quality of the figure 6.
(See page 9, line 327)
Thanking you in advance for your support.
Regards
Seiichi Taguchi
Email: [email protected]